# Temporal dynamics and functional annotation of transcriptome rhythmicity in HEK293T cells

**Yuling Sun[1☯], Huiyu Dong[2☯], Fei Ge[3☯], Ying Zhao[1], Shuhan Yang[1], Yidong Ding[2], Min Dong[3], Liming Wang[3]\*, Tao Zhang[iD][1]\***

**1** Jiangsu Province Engineering Research Center of Development and Translation of Key Technologies for Chronic Disease Prevention and Control, Suzhou Vocational Health College, Suzhou, China, **2** Department of Clinical Medicine, Suzhou Vocational Health College, Suzhou, China, **3** Department of Basic Medicine, Suzhou Vocational Health College, Suzhou, China

☯ These authors contributed equally to this work.
\* zhangtao@szhct.edu.cn (TZ); windywave@163.com (LW)

## Abstract

The endogenous circadian clock drives rhythmic processes in nearly all human cells; however, the temporal organization of the transcriptome in HEK293T cells, a widely used cell line, remains incompletely defined. We synchronized HEK293T cells and performed RNA sequencing at thirteen time points across a 48-hour cycle to map their transcriptome dynamics. Across the time course, principal component analysis revealed clear time point dependent separation of the global transcriptomes; however, coefficient of variation analyses indicated substantially increased divergence among biological replicates starting at T28. In addition, canonical core clock genes showed no detectable circadian rhythmicity when the analysis window extended beyond 28 hours. Genome-wide, only 785 expressed genes displayed rhythmic expression. These rhythmic genes were enriched for cytoplasmic and nuclear compartments, cytoskeletal and membrane related structures, and molecular functions including GTPase activator activity and metal ion binding. Further analysis of expression patterns among arrhythmic genes revealed that only 645 arrhythmic genes displayed time-dependent expression; notably, these genes were enriched in biologically important pathways, including G alpha signaling and structural constituents of chromatin. Together, these results indicate that HEK293T cells exhibit weak intrinsic circadian transcriptome rhythmicity, with most transcripts remaining time independent across the sampled window. This dataset provides a time resolved reference framework to distinguish time-dependent from time-independent gene regulation in HEK293T cells, informing time aware experimental design and interpretation.

## Introduction

The endogenous circadian clock, an evolutionary adaptation to daily environmental cycles, orchestrates rhythmic biological processes. These rhythms arise from

**Data availability statement:** All relevant data are within the manuscript and S5 File The numerical data and summary statistics..All RNA sequencing data files are available from the NCBI GEO database (accession number #GSE315903, https://www.ncbi.nlm.nih.gov/geo/query/acc.cgi?acc=GSE315903).

**Funding:** This work was supported by grants from the Jiangsu Province Engineering Research Center of Development and Translation of Key Technologies for Chronic Disease Prevention and Control (CDSGK12025011, CDSGK12025012), the National Natural Science Foundation of China (32100931 to TZ), the Science and Technology Project of Suzhou (SYWD2024244 to TZ, SYW2025087 to YZ), the General Research Project (Natural Science Project) of Suzhou Vocational Health College (SZWZY202410 to FG). The sponsors or funders play no role in the study design, data collection and analysis, decision to publish, or preparation of the manuscript.

**Competing interests:** The authors have declared that no competing interests exist.

cell-autonomous molecular machinery composed of interlocking transcription-translation feedback loops involving core clock genes expressed in nearly all human cells [1]. In mammals, the transcriptional activators CLOCK and BMAL1 form heterodimers that bind E-box elements and drive the transcription of core repressors, including *PER1*, *PER2*, *PER3*, *CRY1*, and *CRY2*, as well as additional target genes [2]. PER/CRY heterodimers then translocate to the nucleus, inhibiting CLOCK:BMAL1-mediated transcription of their own genes and other circadian target genes [3,4]. Auxiliary loops further stabilize this clockwork. CLOCK:BMAL1 activates transcription of *REV-ERB*s (α, β) and *ROR*s (α, β, γ) [5], which exert opposing effects on *BMAL1* transcription: REV-ERBs repress while RORs activate, creating a critical stabilizing feedback loop [6,7]. Beyond the core network, clock-controlled genes (CCGs), such as *DBP*, fine-tune core loop dynamics and influence oscillatory behavior [8,9].

In mammals, approximately 43% of protein-coding genes exhibit rhythmic expression, with peak times that coordinate diverse biological functions [10]. This temporal regulation operates across multiple layers. At the chromatin level, core clock transcription factors modulate local accessibility and recruit RNA polymerase II, generating genome-wide waves of transcription that follow daily cycles [11,12]. Post-transcriptional mechanisms further refine these rhythms, particularly in RNA processing, where alternative splicing events are temporally regulated by the clock [13,14]. At the translational level, ribosome profiling reveals daily waves of protein synthesis, including increased translation of nuclear-encoded genes that control mitochondrial activity, thereby linking nutrient signals and circadian timing to the liver translatome [2,15].

This temporal orchestration is evident in the regulation of ATP binding proteins, where the circadian clock drives the oscillations of transporters and kinases to synchronize cellular energy expenditure with metabolic demand [16]. Furthermore, the clock modulates GTPase activator activity, providing a mechanism for the time-dependent gating of intracellular signaling cascades and vesicular trafficking essential for cellular homeostasis [17]. Additionally, the rhythmic control of metal ion binding proteins ensures the homeostatic regulation of essential cofactors, such as $Mg^{2+}$ and $Zn^{2+}$, which are critical for both the structural integrity of transcription factors and the enzymatic efficiency of the core molecular oscillator itself [18]. These circadian programs make the timing of experiments a critical factor, as molecular and physiological outcomes vary across the day.

Experiments performed at different circadian phases can produce divergent results. For instance, chromatin immunoprecipitation and related assays capture the time-specific binding of core clock factors and RNA polymerase II recruitment; therefore, ChIP-based measurements depend heavily on the sampling time [19]. Quantitative proteomics reveals daily fluctuations in protein phosphorylation, suggesting that Western blot detection of phosphorylated proteins or kinase activity assays may vary depending on the time of day [20]. Enzymatic assays of nucleotide excision repair also oscillate throughout the day, meaning measurements of repair capacity vary with collection time [21,22]. Additionally, the circadian control of alternative splicing and translational efficiency suggests that RT-PCR splicing analyses, ribosome profiling, or protein synthesis assays may yield non-concordant results if performed at different

time points [23]. These findings highlight the importance of considering circadian timing in experimental design to ensure reproducible and interpretable results.

HEK293T cells originate from human embryonic kidney tissue [24]. The kidney harbors an intrinsic circadian clock that regulates renal solute transport, metabolism, and systemic homeostasis, and it also shapes time-dependent pharmacology [25–27]. In the laboratory, HEK293T cells serve as a versatile platform for molecular and biochemical research. Their high transfection efficiency allows rapid gene overexpression for mechanistic studies and supports the production of correctly folded human proteins for biochemical characterization and structural analysis [28,29]. The line is also widely used for mapping protein interactions via coimmunoprecipitation and affinity purification coupled to mass spectrometry, generating extensive interaction networks [30]. Combined with straightforward culture requirements and robust growth, HEK293T cells provide an ideal system for reporter assays, receptor and signaling studies, as well as other functional cell-based analyses. Despite their widespread use, a comprehensive assessment of circadian transcriptional rhythms in HEK293T cells has been lacking. Establishing the full rhythmic transcriptomic landscape would improve experimental design and enhance the interpretation of temporal data.

In this study, we generated a time-resolved RNA-seq atlas of DEX synchronized HEK293T cells and systematically quantified both circadian and non-circadian transcripts. Despite clear time point dependent separation of global transcriptomes, canonical core clock genes showed no detectable rhythmicity across the T0-T28 window. Using a stringent BH-adjusted Q value threshold (BH.Q < 0.05), we identified circadian rhythmicity in only 4.7% of expressed genes, indicating weak transcriptome-wide oscillation in this model. Extending beyond rhythmicity, we further dissected the arrhythmic transcriptome by adjacent time-point DEG comparisons and found that the vast majority of genes (95.3%) exhibited time-independent expression, consistent with stable baseline transcriptional programs. Notably, the small subset of time-dependent but arrhythmic genes was selectively enriched in biologically important pathways, including G alpha signaling and structure of chromatin. Highlighting these findings, our work provides a practical framework to distinguish time-independent from time-dependent regulation in HEK293T cells, enabling more time-aware experimental design and interpretation.

## Materials and methods

### Cell culture and sample collection

The human embryonic kidney 293T (HEK293T) cell line used in this study was obtained from the American Type Culture Collection (ATCC). Cells were cultured in Dulbecco's Modified Eagle Medium (DMEM) with high glucose, supplemented with 10% fetal bovine serum (FBS), 50 U/mL penicillin, and 50 mg/mL streptomycin, and maintained at 37 °C in 5% $CO_2$. For synchronization, cells were plated in 6-well plates and treated with 200 nM dexamethasone (DEX, Sigma-Aldrich, MO, USA) in PBS for 0.5 hours. Samples were harvested at 4-hour intervals across a 48-hour time course, starting 24 hours after DEX synchronization, yielding a total of 13 time points.

### Transcriptome sequence

Cell pellets were snap frozen and stored at −80 °C. Total RNA was extracted using the TRIzol protocol (Thermo Fisher Scientific). RNA quality and concentration were assessed by RNA electrophoresis, NanoDrop spectrophotometry, and an Agilent 2200 TapeStation. Libraries were prepared and sequenced at the BGI Genome Center (Shenzhen, China) using an Illumina MiSeq platform, which produced paired-end 150 bp reads.

We processed the raw sequencing reads to obtain high-quality clean reads by removing sequencing adapters, short reads (<36 bp), and low-quality reads using Fastp v0.23.2 with non-default parameters (−3 --cut_tail_window_size 4 --cut_tail_mean_quality 20 --detect_adapter_for_pe -l 36). We then assessed read quality with FastQC (default parameters) to confirm data integrity. Clean reads were aligned to the human reference genome (assembly GRCh38) using HISAT2 v2.1.0 with non-default parameters (--rna-strandness RF --dta).

We quantified gene expression as fragments per kilobase of exon per million mapped fragments (FPKM) using String-Tie v1.3.4d with non-default parameters (-e --rf). Differential gene expression was analyzed using the edgeR v3.24.2 R package. We applied the false discovery rate (FDR) method to adjust $P$-values for multiple testing and to determine statistical significance. Only genes with an adjusted $P$-value < 0.05 and $|log_2FC| > 1$ were included in subsequent analyses.

### Rhythmicity assessment and Phase Set Enrichment Analysis (PSEA)

To examine circadian oscillation patterns in transcriptional profiles, we performed rhythmicity analysis using the *meta2d* algorithm (MetaCycle R package, v1.2.0, R Scripts S1 File) [31]. Transcripts with BH.Q < 0.05 were identified as significantly rhythmic; their peak phases were subsequently estimated via sine wave-based least squares fitting (Python Scripts S2 File). We identified peak phases of annotated pathways using Phase Set Enrichment Analysis (PSEA) software [32], which took as input a file containing rhythmic gene names. We obtained updated Gene Ontology (GO) pathway annotations from the Molecular Signatures Database (MSigDB, c5.go.v2024.1.Hs.symbols.gmt) and defined GO terms with Kuiper q-value < 0.05 as significantly enriched (S3 File).

### Functional enrichment analysis

We conducted GO, Kyoto Encyclopedia of Genes and Genomes (KEGG), and Reactome pathway enrichment analyses using the Database for Annotation, Visualization, and Integrated Discovery (DAVID) v6.8. Pathways or terms were considered significantly enriched at an FDR < 0.05.

### Statistics analysis

To compare the distributions of two continuous datasets, we applied a two-sided Kolmogorov–Smirnov (K-S) test. All statistical analyses were conducted using the SciPy library (v1.10.0) in Python, with a significance threshold of 0.05.

## Results

### HEK293T transcriptome profile over 48 hours across 13 time points

To characterize time-of-day effects on the HEK293T transcriptome, we synchronized cultures with dexamethasone for 0.5 h and collected samples every 4h for 24h, starting 24h after synchronization, with three biological replicates per time point. We then performed whole-transcriptome sequencing followed by comprehensive bioinformatic analysis (Fig 1A). We first examined whether time of day shapes global transcriptome organization, principal component analysis (PCA) of transcriptomic profiles across 13 time points revealed clear temporal separation (Fig 1B). Samples progressed sequentially along PC1 and PC2, forming distinct clusters corresponding to successive time points, which reflected the dynamic reorganization of the transcriptome over the circadian cycle, with increased within time point dispersion emerging from T28 onward. Thus, we computed the probability distributions of the FPKM coefficient of variation (CV) within each time point, which showed a pronounced rightward shift toward higher CV values after T28 (Fig 1C). We next used MetaCycle to assess gene expression rhythmicity across different analysis window lengths and found that, whether using nominal P values or the more stringent BH-adjusted Q values, the number of rhythmic genes dropped sharply once the time window was extended to 32 h (Fig 1D).

Taken together, the HEK293T transcriptome exhibited clear time point dependent differences, suggesting potential circadian regulation. Balancing the need for a time window longer than 24 h with reproducibility across biological replicates, we therefore selected T0 to T28 as the analysis window and applied the more stringent BH. Q value to define rhythmic transcripts for subsequent analyses.

### Temporal expression patterns of core circadian genes from T0 to T28

Because core clock oscillations primarily determinant of transcriptome-wide rhythmicity, we first characterized the temporal expression patterns of core clock genes across the T0 to T28 period (Fig 2A). Fig 2 summarizes the expression of

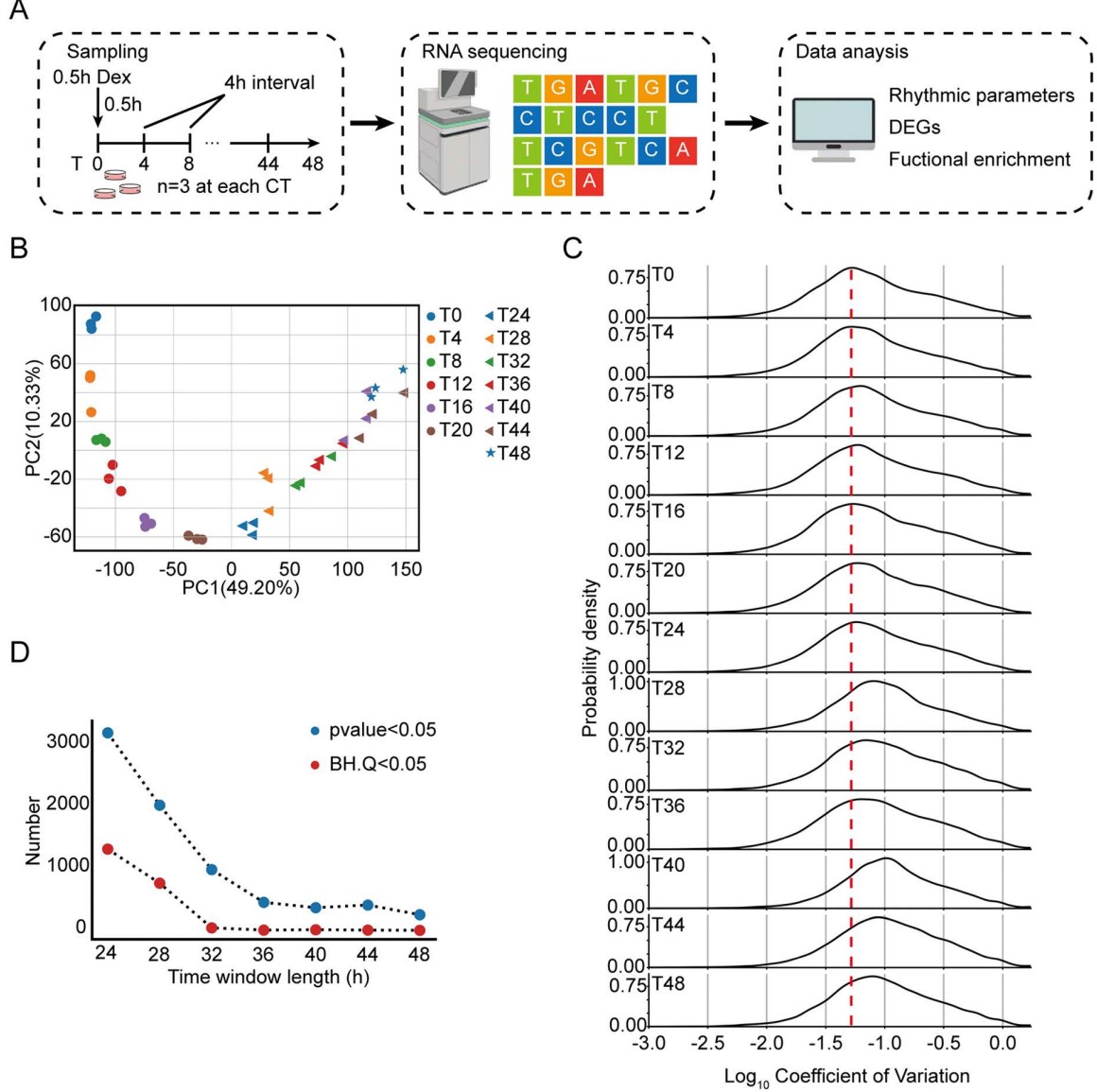

**Fig 1. Experimental design and global identification of rhythmic transcripts.** (A) Time course RNA sequencing profiles and analyses across 28 hours from T0 to T28 at 4-hour intervals, initiated 24 hours after DEX synchronization. (B) Principal component analysis (PCA) of RNA-seq samples. Points are colored by time. Axes display PC1 and PC2 with the percentage of variance explained. (C) The distribution of the coefficient of variation (CV) of gene expression across three biological replicates at each time point, and the red line indicates the mean CV at T0. Kolmogorov–Smirnov (K-S) test $P < 0.001$ (T0 v.s. T28, T32, T36, T40, T44, T48). (D) The number of rhythmic genes identified across different analysis window lengths using the meta2d algorithm, based on thresholds of P value < 0.05 or BH-adjusted Q value < 0.05.

ten core circadian clock genes and two clock output genes over 28 hours. When rhythmicity was assessed using the data between T0 and T24, *BMAL1* exhibited detectable circadian rhythmicity, whereas no significant rhythmicity was detected for the remaining genes. In contrast, analysis of the full 28 hours failed to identify significant rhythmicity for any of the ten core clock genes. *PER1*, *CRY1*, *NR1D1*, and *RORA* displayed a progressive upward trend, whereas *CLOCK* and *PER2* showed a gradual decline. In contrast, *BMAL1*, *PER3*, and *CRY2* remained comparatively stable with limited temporal

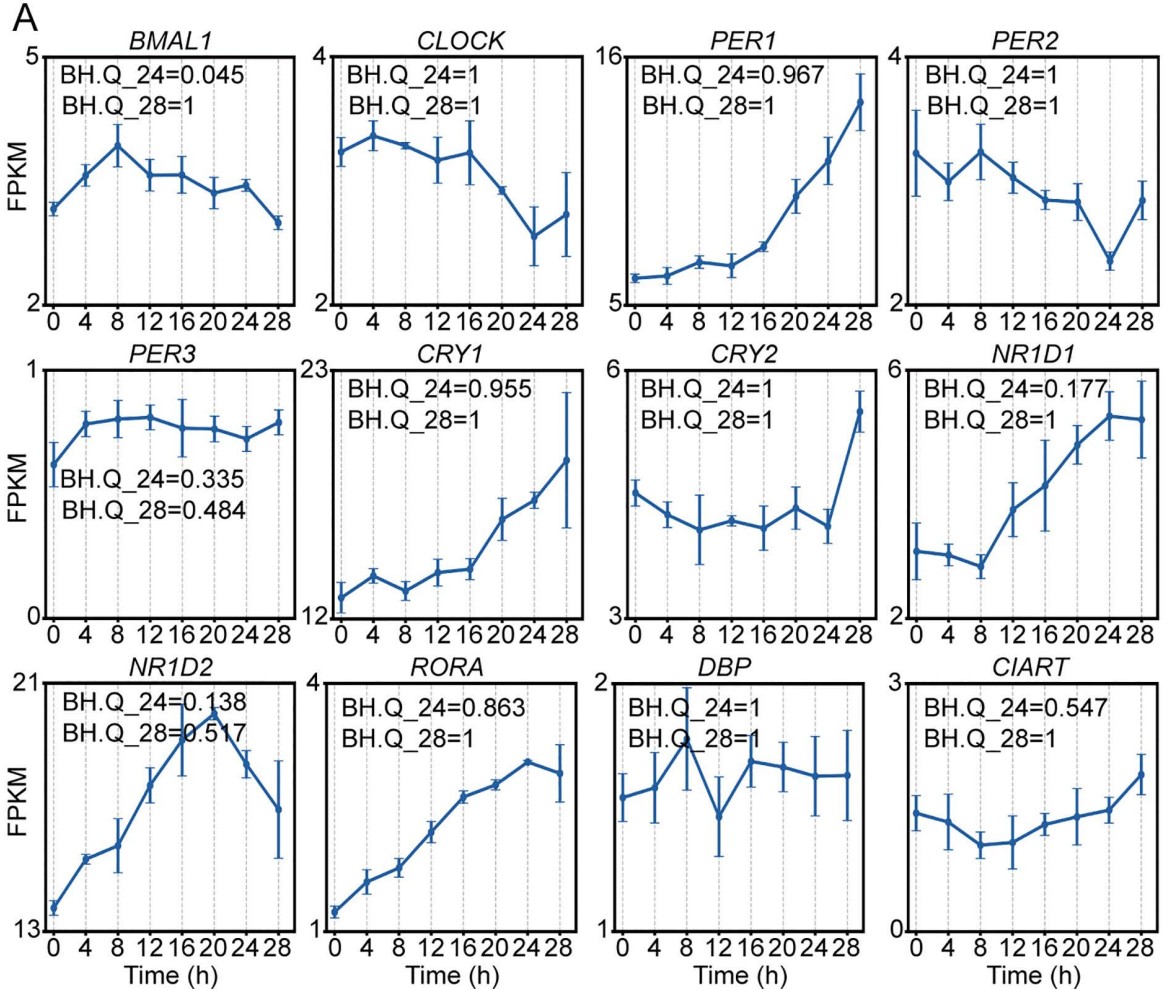

**Fig 2. Core clock gene expression in HEK293T cells across the T0-T28.** (A) Time-series expression profiles of 10 core clock genes in HEK293T cells. Expression values represent mean FPKM±SD, n = 3. The BH. Q values calculated from the 24 h and 28 h datasets using the meta2d algorithm are annotated in each panel.

variation. *NR1D2* exhibited a pronounced peak at T20, which may reflect an asymmetric expression trajectory, yet it still did not reach significance for circadian rhythmicity. Among the two clock output genes, *DBP* showed modest undulations superimposed on an otherwise steady baseline, while *CIART* remained largely constant across the time course.

Overall, core clock genes did not exhibit significant circadian rhythmicity across the 28h time course, suggesting that transcriptome wide oscillations are weak in HEK293T cells under these conditions. Consistently, PCA of the transcriptomic profiles showed that samples post T24 remained markedly separated from earlier time points rather than progressively converging toward the cluster of T0 to T20, further supporting attenuated global rhythmicity.

## Time of day structures the HEK293T transcriptome

Given that none of the core clock genes exhibited detectable circadian rhythmicity in HEK293T cells, we next extended our analysis to the transcriptome level to systematically assess rhythmicity across the HEK293T transcriptome. Circadian rhythm analysis using MetaCycle identified that 785 rhythmic genes (4.7%), whereas the remaining 16,064 genes (95.3%)

were classified as arrhythmic (Fig 3A). To characterize the expression phase of these rhythmic genes, we estimated their phases using a sine wave based least squares fitting approach. Fig 3B illustrates that the phases of these rhythmic genes were predominantly concentrated between T2 and T16, with a smaller subset showing clustered expression at the anti phase window of T0 to T5. To determine which pathways these rhythmic genes are associated with and what biological processes they may regulate, we performed GO, KEGG, and Reactome enrichment analyses on the 785 rhythmic genes. The enrichment results (Fig 3C) showed that these rhythmic genes were mainly associated with cytoplasmic and nuclear compartments (cytoplasm, nucleoplasm, cytosol), cytoskeletal and cellular structural components (cytoskeleton, cell projection, endomembrane system), and junction or membrane related terms (anchoring junction, basement membrane, transport vesicle membrane). In terms of molecular function, prominent categories included protein binding and nucle-otide binding, notably ATP binding, GTPase activator activity, and metal ion binding (including zinc ion binding). KEGG and REACTOME analyses further highlighted aldosterone synthesis and secretion, IgSF CAM signaling, and choles-terol biosynthesis. Within the GTPase activator activity pathway, the representative genes ACAP2 and DOCK1 exhibited expression peaks at T14.69 and T12.94, respectively, whereas RANGAP1 displayed a trough around T12. Similarly, the representative genes MICU1, CETN3, and COX5B in the zinc ion binding pathway showed a comparable temporal pattern (Fig 3D).

To define biologically coherent gene sets with temporally coordinated expression in HEK293T cells, we applied PSEA to the phase distribution of rhythmically expressed genes (S4 File). Because the peak phases of rhythmic gene expression were relatively clustered, the PSEA results encompassed a broad range of significantly enriched pathway terms, with phases largely concentrated between T12 and T14, including basement membrane and transport vesicle membrane (Fig 3E).

Overall, these analyses indicate that only a small fraction of the expressed genes in HEK293T cells exhibit circadian rhythmicity, with peak expression of rhythmic transcripts largely clustered between T12 and T16, and that the rhythmic genes are primarily enriched in pathways and functions such as GTPase activator activity and metal ion binding.

## Time-dependent and time-independent expression patterns revealed by DEG analysis of arrhythmic genes

In the analyses described above, only 4.7% of expressed genes in HEK293T cells exhibited detectable circadian rhyth-micity. However, incorporating time as an experimental variable revealed that transcriptional dynamics are not limited to rhythmic oscillations alone. To further characterize temporal regulation among arrhythmic genes, we classified genes based on their differential expression behavior between adjacent time points from T0 to T2. Genes that showed no differ-ential expression in any pairwise comparison between adjacent time points were defined as time-independent, whereas genes that were differentially expressed in at least one adjacent time-point comparison were classified as time-dependent, despite lacking detectable rhythmicity (Fig 4A). Based on this criterion, we performed pairwise DEG analyses between all neighboring time points across the T0 to T28 interval, focusing exclusively on arrhythmic genes.

Using this approach, we identified 15,419 time-independent genes whose expression levels remained stable across all adjacent time-point comparisons (Fig 4B). Functional enrichment analysis revealed that these genes were predominantly associated with fundamental cellular processes, including mitotic spindle assembly checkpoint signaling and protein tar-geting to mitochondrion (BP), components of the U2-type precatalytic spliceosome and the U12-type spliceosomal com-plex (CC), as well as molecular functions such as histone H3 methyltransferase activity and single-stranded DNA helicase activity (MF). In addition, pathway-level annotations highlighted RNA polymerase and autophagy (KEGG), together with vesicular trafficking–related processes such as the Endosomal Sorting Complex Required For Transport (ESCRT) and budding and maturation of HIV virion (Reactome) (Fig 4C).

In contrast, we identified 645 time-dependent but arrhythmic genes, which exhibited differential expression in at least one adjacent time-point comparison (Fig 4D). Notably, the overlap of DEGs between different adjacent time point com-parisons was minimal, indicating that time dependent expression changes were highly specific to individual temporal

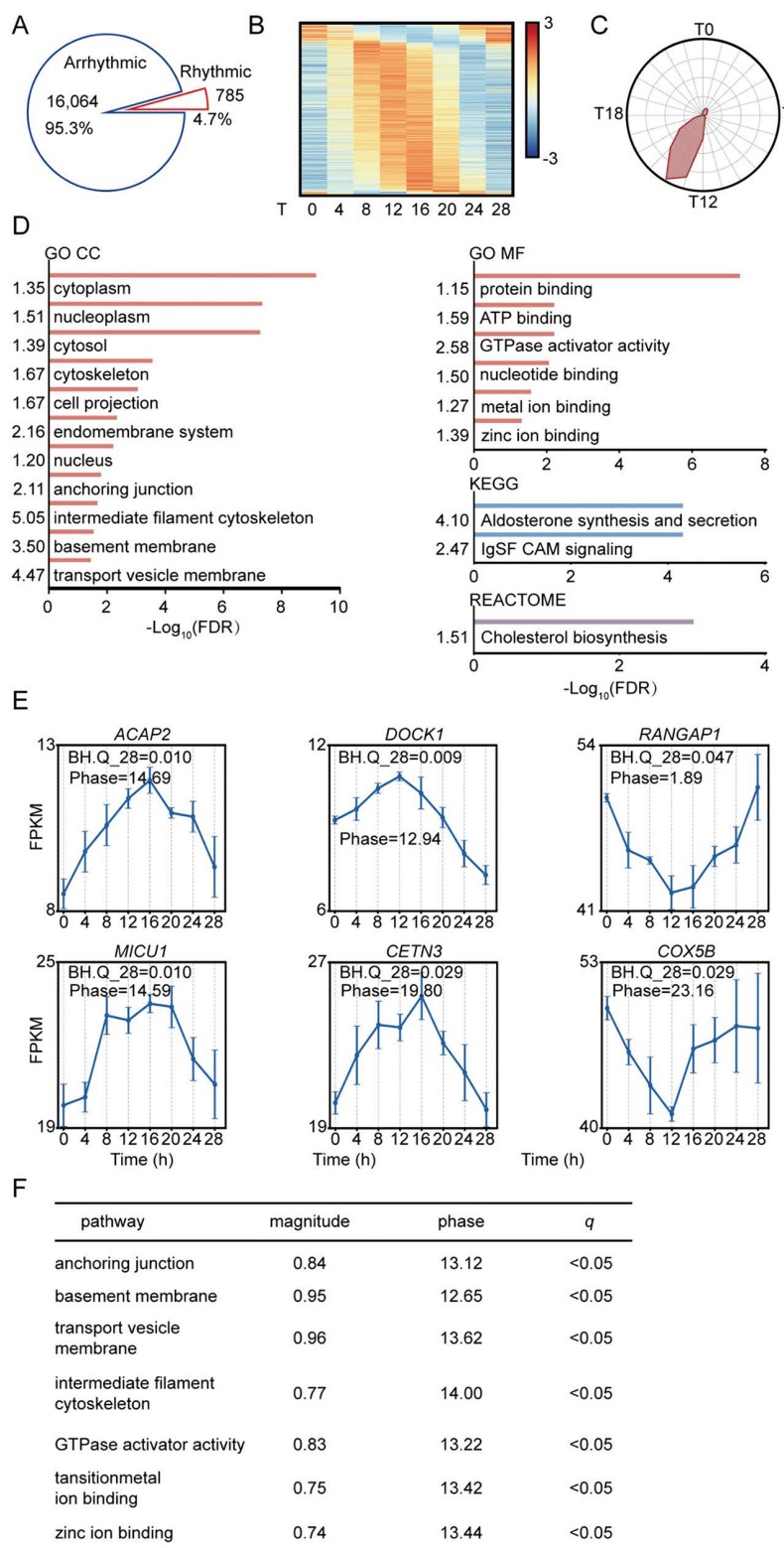

**Fig 3. Circadian features of the HEK293T cell transcriptome.** (A) Proportion of rhythmic and arrhythmic genes among all detected genes. Counts and percentages are shown. (B) Heatmaps displaying rhythmic genes HEK293T cells. Genes are ordered by peak phase in the rhythmic condition.

(C) Phase distribution of rhythmic genes across the 24-hour cycle. (D) GO, KEGG and REACTOME pathway enrichment analyses of rhythmic genes. (E) Representative GTPase activator and metal ion binding genes. Expression values represent mean FPKM±SD, n = 3. BH.Q are calculated using the meta2d algorithm, Phases were estimated using sine wave–based least-squares fitting. (F) Pathway terms consistent between PSEA and pathway enrichment analysis.

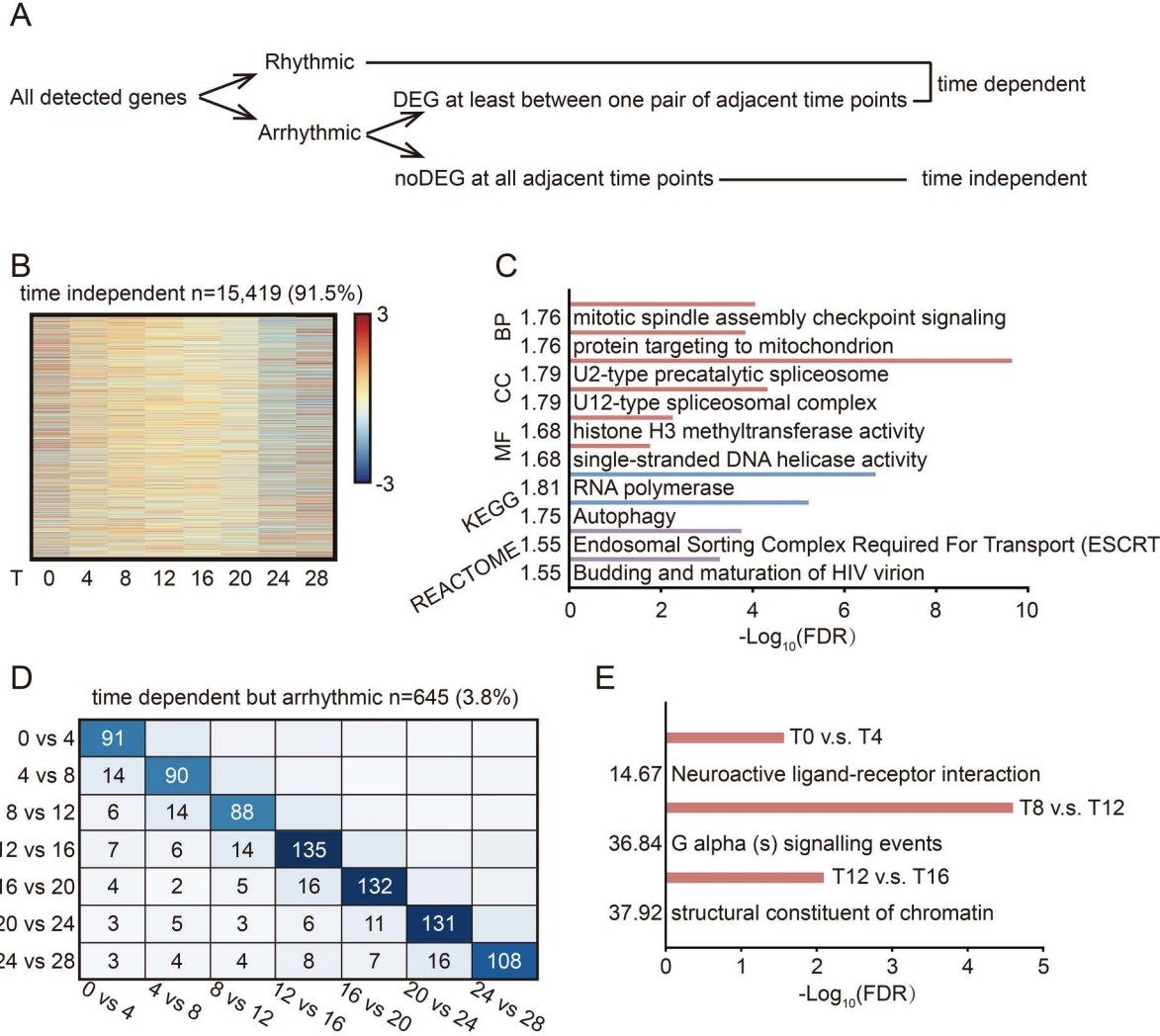

**Fig 4. Classification of time-dependent and time-independent expression patterns among arrhythmic genes.** (A) Schematic overview of the analytical framework used to classify arrhythmic genes into time-independent and time-dependent categories based on differential expression between adjacent time points from T0 to T28. (B) Heatmaps displaying time independent genes HEK293T cells. (C) GO, KEGG and REACTOME pathway enrichment analyses of time independent genes. (D) The number of overlapping genes between DEG sets. (E) Pathway enrichment analyses of time dependent genes in three DEG group: T0 v.s. T4, T8 v.s. T12, T12 v.s. T16.

transitions rather than shared across multiple intervals. This observation suggests that these genes respond to discrete time associated regulatory cues rather than sustained or oscillatory regulation.

Due to the relatively small number of DEGs detected in each adjacent time-point comparison, pathway enrichment analysis yielded statistically significant results (FDR < 0.05) in only three DEG sets, namely Neuroactive

ligand-receptor interaction (T0 vs. T4), G alpha (s) signaling events (T8 vs. T12), and structural constituent of chromatin (T12 vs. T16) (Fig 4E).

Collectively, these results suggest that the majority of genes in HEK293T cells are expressed in a time-independent manner and are enriched in a broad spectrum of essential biological processes.

## Discussion

We profiled the temporal transcriptome of dexamethasone-synchronized HEK293T cells and found clear time-dependent expression changes but weak circadian rhythmicity. Core clock genes lacked robust oscillation beyond 28 h, and only a small gene subset showed rhythmic expression, enriched in GTPase activator activity and metal ion binding, peaking mainly between T12 and T16. Beyond rhythmic transcripts, we further stratified the arrhythmic transcriptome by adjacent time-point differential expression and found that most genes were time independent, whereas a small subset of time-dependent but arrhythmic genes was enriched in key pathways, including G alpha signaling events and structural constituents of chromatin. Collectively, these data indicate that most genes in HEK293T cells are expressed in a largely time independent manner over the sampled window, while a small, pathway focused subset shows discrete time-dependent regulation that should be considered when designing and interpreting HEK293T-based experiments.

While our findings highlight the potential importance of time of day in HEK293T cell experiments, we acknowledge that under normal conditions, cells are not as strictly synchronized as they would be with dexamethasone treatment or serum shock. For many experimental contexts, such synchronization may not be feasible or desirable. Although circadian rhythmicity is weak at the transcriptome-wide level in HEK293T cells, time-dependent gene expression patterns are still significant, suggesting that time of day should be considered in experimental design even without strict phase control.

In consideration of acute DEX-induced transcriptional bursts, we initiated collection 24 hours post-synchronization, covering the 48 hours period. This 24-hour delay is crucial as it allows the transient, GR-mediated transcriptional burst, typically characterized by immediate early gene induction. By focusing on the 48-hour interval, the observed mRNA fluctuations are more likely to reflect stabilized, clock-driven endogenous rhythms rather than acute pharmacological responses to the initial DEX stimulus. To more rigorously decouple these effects, future studies should include a non-synchronizing control or utilize inhibitors of GR signaling to isolate clock-dependent outputs. Furthermore, our 4-hour resolution across this stabilized second and third cycle provides a more rigorous basis for calculating phase and amplitude. This temporal separation effectively decouples the acute chemical signaling from the circadian output, addressing the ambiguity inherent in early-cycle sampling.

Previous investigations into HEK293T cells observed that while PER2 exhibited moderate oscillations 24 hours post-DEX synchronization, CLOCK failed to show a discernible rhythmic pattern, and both lacked robust statistical support [33]. In contrast, our data demonstrate significantly reduced variance among biological replicates within the 24–48-hour window, thereby enhancing the statistical power and reliability of our conclusions. We propose that these divergent expression patterns are a result of our optimized synchronization protocol; by shortening the DEX exposure from 2 hours to 0.5 hours, we effectively minimized exogenous interference, allowing for a more accurate reflection of endogenous circadian dynamics.

Overall, these data provide a framework for time-aware experimentation while also exposing gaps in sampling, modality, and causality that future studies should address. Bulk RNA-seq cannot resolve cell-state heterogeneity or post-transcriptional regulation; integrating ribosome profiling, quantitative proteomics, and phosphoproteomics will be crucial in determining whether phase-ordered transcriptomes extend to translation and signaling. Finally, generalization beyond HEK293T requires testing additional human cell types and primary cells under physiologically relevant zeitgebers.

Future studies should extend these findings across multiple circadian cycles and integrate multi-omics approaches to track how phase information propagates from chromatin to proteins. Implementing live reporters and CRISPR-based

perturbations of core clock components will be essential to establish causal links between circadian phase and pathway activity. Additionally, real-time monitoring of membrane trafficking and ciliary function will help connect temporal transcriptional programs to cellular behaviors. These insights can be translated into practical guidance for experiments in HEK293T cells by defining fixed sampling windows, verifying phase using reporters, and incorporating circadian phase as a covariate in all analyses.

## Conclusion

We generated a time-resolved RNA-seq atlas of DEX-synchronized HEK293T cells and observed clear time point dependent transcriptome separation, with reduced reproducibility after T28 supporting a T0-T28 focus. Core clock genes showed limited rhythmicity, and only 785 genes (4.7%) were rhythmic, with phases largely clustered at T12–T16 and enrichment in GTPase activator and metal ion binding functions. Among arrhythmic genes, most were time independent, whereas 645 were time dependent and enriched in Gα signaling and chromatin-related processes. This resource supports time-aware experimental design in HEK293T cells.

## Supporting information

**S1 File. R scripts for MetaCycle rhythmicity analysis.**
(DOCX)

**S2 File. Python scripts for sine-wave-based estimation.**
(DOCX)

**S3 File. PSEA analysis.**
(DOCX)

**S4 File. PSEA results.**
(XLSX)

**S5 File. The numerical data and summary statistics.**
(XLSX)

## Acknowledgments

We thank Songbai Liu from Suzhou Vocational Health College for providing cell materials and offering constructive suggestions.

## Author contributions

**Conceptualization:** Liming Wang, Tao Zhang.

**Data curation:** Yuling Sun, Huiyu Dong, Fei Ge, Ying Zhao, Shuhan Yang, Yidong Ding, Min Dong, Liming Wang, Tao Zhang.

**Formal analysis:** Yuling Sun, Huiyu Dong, Fei Ge, Ying Zhao, Shuhan Yang, Liming Wang, Tao Zhang.

**Funding acquisition:** Fei Ge, Ying Zhao, Tao Zhang.

**Investigation:** Yuling Sun, Huiyu Dong, Fei Ge, Ying Zhao, Liming Wang, Tao Zhang.

**Project administration:** Yidong Ding, Min Dong, Tao Zhang.

**Resources:** Liming Wang.

**Supervision:** Tao Zhang.

**Writing – original draft:** Yuling Sun, Fei Ge, Liming Wang, Tao Zhang.

**Writing – review & editing:** Yuling Sun, Huiyu Dong, Fei Ge, Ying Zhao, Shuhan Yang, Yidong Ding, Min Dong, Liming Wang, Tao Zhang.

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
