## [Decision Letter · Decision Letter 0]

8 Dec 2025

Dear Dr. Zhang,

Thank you for submitting your manuscript to PLOS ONE. After careful consideration, we feel that it has merit but does not fully meet PLOS ONE’s publication criteria as it currently stands. Therefore, we invite you to submit a revised version of the manuscript that addresses the points raised during the review process.

It is particularly important to address the concern that relates to the experimental design used for this study. Sampling was performed over a 24 hours period and was initiated immediately following the proposed clock synchronisation treatment. Therefore, it is currently not possible to conclude that the observed dynamic changes in gene expression relate to a rhythmic output of the circadian clock. It cannot be excluded that these changes reflect acute responses to the cell culture treatment. This point needs to be resolved by a longer sampling period that is delayed (by 24 hours) relative to the Dex treatment. Other critical points raised by both reviewers also need to be addressed.

We look forward to receiving your revised manuscript.

Kind regards,

Nicholas S. Foulkes, D.Phil

Academic Editor

PLOS One

Journal Requirements:

4. We note that your Data Availability Statement is currently as follows: “All relevant data are within the manuscript and its Supporting Information files.All RNA sequencing data files are available from the NCBI GEO database (accession number #GSE308811, https://www.ncbi.nlm.nih.gov/geo/query/acc.cgi?acc=GSE308811).”

Additional Editor Comments:

Major revision is required, in particular in order to support the conclusion that these rhythmic elements of the transcriptome reflect circadian clock functionality. Currently the sampling period is too short (24 hours) and the sampling starts immediate following Dex treatment. For this reason it is not possible to assess whether the observed changes reflect the consequences of the acute exposure to Dex exposure or whether they are clock regulated. The other comments made by both reviewers also need to be addressed in full.

Reviewers' comments:

Reviewer's Responses to Questions

**Comments to the Author**

1. Is the manuscript technically sound, and do the data support the conclusions?

Reviewer #1: Yes

Reviewer #2: No

2. Has the statistical analysis been performed appropriately and rigorously?

Reviewer #1: Yes

Reviewer #2: No

3. Have the authors made all data underlying the findings in their manuscript fully available?

Reviewer #1: No

Reviewer #2: No

4. Is the manuscript presented in an intelligible fashion and written in standard English?

Reviewer #1: Yes

Reviewer #2: Yes

Reviewer #1: In their manuscript “Temporal dynamics and functional annotation of transcriptome rhythmicity in HEK293T cells”, the authors examine RNA expression patterns at six time points across 24 hours after synchronization of the cells by dexamethasone treatment. They identify enrichment of certain functions, such as mitochondrial translation and RNA splicing, in genes clustered into distinct phases, and observe that gene expression peaks progress from genes related to transcriptional control to those related to translation across the 24 h time period. When looking at differentially expressed genes between antiphase time points, they notice that most genes underlying these differences are rhythmically expressed based on MetaCycle analysis, and show enriched pathway functions such as vesicle trafficking or fatty acid metabolism.

Data quality and analysis are sound, and the conclusions taken by the authors are valid. However, I have a few points that should be addressed prior to publication.

Major points:

A) The authors follow gene expression over 24 h only, thus only one circadian cycle. This means that their gene set will contain genes rhythmically expressed after DEX treatment, but also genes acutely induced by the treatment, especially in the first few hours after the treatment. For a more stringent identification of oscillating genes, the authors would have to follow expression over at least two cycles or 48 hours. The discussion section devoted to these issues (acute DEX-mediated induction of expression vs. rhythmic expression in response to core clock synchronization) should be expanded (Currently only: “Our 4-hour resolution sampling across 24 hours captures one cycle; extending to multiple cycles with denser sampling would refine phase and amplitude estimates and increase sensitivity for low-amplitude rhythms.” It would also help to distinguish acute effects of the DEX treatment from those brought about by synchronization of the clock, for example).

B) The RNA sequencing data should be deposited in GEO (https://www.ncbi.nlm.nih.gov/geo/) or a similar public repository supporting MIAME-compliant data submissions. Please provide an accession number upon revision.

Minor points:

1) “In mammals, approximately 43% of protein-coding genes exhibit rhythmic expression…”: The cited reference (No. 6) does not contain this information. Please provide a reference for this statement.

2) “CLOCK peaked 32–36 hours after dexamethasone synchronization, in agreement with previous reports (25).” This statement is in contradiction with the rest of the manuscript, where expression was followed only for 24 h after DEX treatment.

3) Figure 2 A: were there only two samples for CT12, or is there a close overlap of two samples?

4) “…and translation-related enzymes (MFMT, QRSL1) (Fig. 2E).” This should refer to panel 2F.

5) “…peaked between CT10 and CT12 (Fig. 2F),…” This should refer to panel 2G.

6) “Collectively, these findings establish time of day as a critical experimental variable in HEK293T cells and highlight the need for either strict phase control or phase-aware analyses to ensure reproducibility and accurate interpretation.” This is correct; however, under general culture conditions, cells will not be as strictly synchronized as upon DEX treatment or serum shock. It is thus not easy to establish phase control or phase awareness in praxis (unless one wants to always synchronize cells by DEX/serum shock, which represents a relatively harsh treatment in itself that may not be compatible with many other treatments). The authors should expand the discussion here a bit to reflect on these practical problems.

7) “At CT16, vesicular transport genes were strongly induced, including DNM1 (clathrin-mediated endocytosis), GBF1 (ARF1 activation and Golgi–ER recycling), SEC24A (COPII cargo selection), and HGS/VPS36 (ESCRT sorting), suggesting circadian control of membrane trafficking and receptor signaling capacity, such as glucocorticoid signaling (45).” As the glucocorticoid receptor is a nuclear receptor and not a membrane receptor, this statement is a bit confusing. Also, the cited reference only comments on membrane vesicles in the context of adrenocortical cells and increased release of steroids from these cells under stress, which is not applicable to the HEK239T cells. Please clarify and/or change references.

Reviewer #2: The study by Zhang and colleagues addressed an important question regarding how HEK293 endogenous rhythms function. The authors highlighted the consequences of not considering time in experiments, which is highly relevant to the field. However, upon careful review of the methods, the current rhythm analyses are underpowered and susceptible to a high rate of false positives. The authors used uncorrected p-values to determine rhythmicity, resulting in a large number of rhythmic genes—close to half of the transcriptome. Based on this data, they also performed phase set enrichment analyses (PSEA), which similarly used uncorrected p-values. Overall, the manuscript’s conclusions are not supported by robust statistical analysis. Although the manuscript is well-written and presents a good idea, the conclusions drawn are insufficient. The authors should reanalyze their data while accounting for false discovery rates, consider using sine-wave-based methods to accurately estimate phase, and then perform PSEA. Additionally, they could use DEG analyses to identify truly arrhythmic processes, providing a balanced view of major processes that are either under temporal control or not. This could become a valuable resource for the scientific community. I also provide some other comments below.

Introduction

Add original studies to support the claims. For instance, reference 6 is a review. The authors should cite the original paper.

M&M

Usually, after dexamethasone synchronization, one should wait 24 hours to avoid the immediate GR-induced effects. In this setup, the authors proceed immediately after synchronization, which should be disclosed as a limitation. Additionally, the authors could select a few relevant genes and repeat the experiments, this time waiting 24 hours, to show that the induced effect is clock-mediated rather than GR-induced.

The authors used only the metacycle package to estimate rhythmicity, which includes ARSER, JTK_Cycle, Lomb-Scargle, and meta2d. In this setup, it is likely that Meta2D incorporated JTK_Cycle and Lomb-Scargle methods. An important limitation of this approach is that these methods are not sine-wave based, making phase and amplitude estimation difficult. Specifically, the authors used the calculated phase for PSEA analysis. Importantly, using a sine-wave method in addition could help obtain high-resolution phase estimates, although the overlap between detection methods is often quite limited.

In addition, all analyses were based on a standard p-value < 0.05, which, considering RNAseq data, could inflate the rhythm detection. In this case, a false-discovery rate is necessary. Since some of these methods have low statistical power, using a p-value < 0.01 could help reduce false positives. Not surprisingly, the authors report almost half of the transcriptome as rhythmic, which is likely due to the accumulation of false positives.

The same issue applies to the PSEA analysis, as the authors used a p-value < 0.05. Here, a q-value should be used instead of < 0.05 to reduce false positives. Data processing for PSEA is unclear and not reproducible. This method produces a large number of biological processes, often overlapping. How was the merging or grouping of similar processes done? Based on the data shown in fig. 2I, this reviewer cannot replicate these findings with the information provided by the authors. Importantly, PSEA data is not included in the materials; this must be provided.

Results

The authors should consider using a sine-wave baseline method for the clock genes to better assess the classic anti-phase relationship. Surprisingly, DBP, a high-amplitude gene, is not rhythmic.

The phase shown in D could be better represented with a rose plot.

The results from Fig. 2J need to be reanalyzed, accounting for false discoveries as described above.

It is confusing to evaluate rhythmicity using a combination of methods and then perform DEG analyses, which is not suitable for circadian studies. In this current format, I believe this type of analysis does not contribute the story and instead introduces additional confusion. Although the authors attempted to evaluate the potential overlap between PSEA and DEG analyses, it remains unclear how similar or different the outcomes are. An important aspect missing from this paper is focusing on the arrhythmic portion. For example, the authors could assess rhythmicity—now accounting for false discovery rates—and perform DEG analysis specifically within the arrhythmic part of the transcriptome. This approach would identify the main genes and pathways that are time-independent. Subsequently, they could conduct rhythm analysis to identify genes and pathways that are time-dependent. Comparing these results would allow the authors to suggest a list of processes that are either time-dependent or time-independent.

**Do you want your identity to be public for this peer review?** For information about this choice, including consent withdrawal, please see our Privacy Policy

Reviewer #1: No

Reviewer #2: No

---

## [Author Response · Author response to Decision Letter 1]

3 Feb 2026

Dear Foulkes,

Thank you very much for your reply and for giving us chance to revise our manuscript according to the reviewers’ and your constructive comments. We are pleased that you and the reviewers consider our study of sufficient interest for possible publication in PLOS ONE. We have taken the comments on board to improve and clarify the manuscript. Please find below a detailed point-by-point response to all comments (academic editor and reviewers’ comments in blue, our responses in black).

Academic editor

Major revision is required, in particular in order to support the conclusion that these rhythmic elements of the transcriptome reflect circadian clock functionality. Currently the sampling period is too short (24 hours) and the sampling starts immediately following Dex treatment. For this reason, it is not possible to assess whether the observed changes reflect the consequences of the acute exposure to Dex exposure or whether they are clock regulated. The other comments made by both reviewers also need to be addressed in full.

Response: We thank the editor for pointing this out and apologize for the experimental design mistake. We redesigned the experiment as follows: HEK293T cells were synchronized with dexamethasone for 0.5 h, washed with PBS, and replenished with fresh dexamethasone-free complete medium. Starting 24h after synchronization, samples were collected every 4h over a 48h window, with three biological replicates at each time point (Line 132-136). The process of sample collection and data annlysis is shown in Fig 1A. Collected samples were subjected to RNA sequencing the raw data have already been uploaded to GEO. The GEO accession is GSE315903 (https://www.ncbi.nlm.nih.gov/geo/query/acc.cgi?acc=GSE315903). If needed, it can be accessed using the token qzibekaajlqbxan.

Fig 1A

Reviewer #1:

In their manuscript “Temporal dynamics and functional annotation of transcriptome rhythmicity in HEK293T cells”, the authors examine RNA expression patterns at six time points across 24 hours after synchronization of the cells by dexamethasone treatment. They identify enrichment of certain functions, such as mitochondrial translation and RNA splicing, in genes clustered into distinct phases, and observe that gene expression peaks progress from genes related to transcriptional control to those related to translation across the 24 h time period. When looking at differentially expressed genes between antiphase time points, they notice that most genes underlying these differences are rhythmically expressed based on MetaCycle analysis, and show enriched pathway functions such as vesicle trafficking or fatty acid metabolism.

Data quality and analysis are sound, and the conclusions taken by the authors are valid. However, I have a few points that should be addressed prior to publication.

Response: We sincerely appreciate the reviewer’s positive evaluation and constructive comments, which have greatly contributed to the refinement of our study.

Major points:

A) The authors follow gene expression over 24 h only, thus only one circadian cycle. This means that their gene set will contain genes rhythmically expressed after DEX treatment, but also genes acutely induced by the treatment, especially in the first few hours after the treatment. For a more stringent identification of oscillating genes, the authors would have to follow expression over at least two cycles or 48 hours.

Response: We thank the reviewer for pointing this out and apologize for the experimental design mistake. As both the Editor and the Reviewer #2 also pointed out, we have redesigned the experiment to provide a more comprehensive temporal analysis. Specifically, we collected samples at 13 time points over a 48-hour period, with sampling initiated 24 hours after DEX synchronization (Line 132-136). The process of sample collection and data analysis is shown in Fig 1A.

The discussion section devoted to these issues (acute DEX-mediated induction of expression vs. rhythmic expression in response to core clock synchronization) should be expanded (Currently only: “Our 4-hour resolution sampling across 24 hours captures one cycle; extending to multiple cycles with denser sampling would refine phase and amplitude estimates and increase sensitivity for low-amplitude rhythms.” It would also help to distinguish acute effects of the DEX treatment from those brought about by synchronization of the clock, for example).

Response: We thank the reviewer for this insightful comment. We agree that distinguishing acute DEX-mediated induction from clock-driven rhythms is essential. Consequently, in our revised study, we have shifted the sampling window to start from 24 hours post-DEX treatment. We have incorporated a detailed discussion regarding the superposition of acute DEX effects and circadian oscillations, clarifying how our sampling strategy (24–72 h post-treatment) effectively isolates the latter (Line 327-339).

B) The RNA sequencing data should be deposited in GEO (https://www.ncbi.nlm.nih.gov/geo/) or a similar public repository supporting MIAME-compliant data submissions. Please provide an accession number upon revision.

Response: We thank the reviewer for pointing out this. The raw data of RNA sequencing have already been uploaded to GEO. The GEO accession is GSE315903 (https://www.ncbi.nlm.nih.gov/geo/query/acc.cgi?acc=GSE315903). If needed, it can be accessed using the token qzibekaajlqbxan.

Minor points:

1) “In mammals, approximately 43% of protein-coding genes exhibit rhythmic expression…”: The cited reference (No. 6) does not contain this information. Please provide a reference for this statement.

Response: We appreciate the reviewer for raising the question and sorry for not citing corresponding reference. We have incorporated the appropriate reference in the revised manuscript.

2) “CLOCK peaked 32–36 hours after dexamethasone synchronization, in agreement with previous reports (25).” This statement is in contradiction with the rest of the manuscript, where expression was followed only for 24 h after DEX treatment.

Response: The reviewer is quite right. In the revised manuscript, we redesigned the experiments and collected new samples. The current results indicate that the expression of CLOCK exhibits a progressive downward trend rather than a distinct circadian rhythm. Consequently, the original statement has been removed. We have addressed the difference between our observed expression patterns and those reported in previous studies in the Discussion section.

3) Figure 2 A: were there only two samples for CT12, or is there a close overlap of two samples?

Response: We appreciate the reviewer’s careful observation. It is indeed a close overlap of two samples at CT12 in the initial version of Fig 2A. A similar situation occurs in the revised Figure 1. To ensure full transparency and clarity, we have summarized the raw data and summary statistics for all figures in S5 File, which has been uploaded as Supplementary Material.

4) “…and translation-related enzymes (MFMT, QRSL1) (Fig. 2E).” This should refer to panel 2F.

Response: Due to the new experimental design and updated analytical results, this specific result and the corresponding figure have been removed in the revised manuscript.

5) “…peaked between CT10 and CT12 (Fig. 2F),…” This should refer to panel 2G.

Response: Due to the new experimental design and updated analytical results, this specific result and the corresponding figure have been removed in the revised manuscript.

6) “Collectively, these findings establish time of day as a critical experimental variable in HEK293T cells and highlight the need for either strict phase control or phase-aware analyses to ensure reproducibility and accurate interpretation.” This is correct; however, under general culture conditions, cells will not be as strictly synchronized as upon DEX treatment or serum shock. It is thus not easy to establish phase control or phase awareness in praxis (unless one wants to always synchronize cells by DEX/serum shock, which represents a relatively harsh treatment in itself that may not be compatible with many other treatments). The authors should expand the discussion here a bit to reflect on these practical problems.

Response: We thank the reviewer for the insightful comments. As suggested by the reviewer we have expanded our Discussion to address these practical issues (Line 319-326).

7) “At CT16, vesicular transport genes were strongly induced, including DNM1 (clathrin-mediated endocytosis), GBF1 (ARF1 activation and Golgi–ER recycling), SEC24A (COPII cargo selection), and HGS/VPS36 (ESCRT sorting), suggesting circadian control of membrane trafficking and receptor signaling capacity, such as glucocorticoid signaling (45).” As the glucocorticoid receptor is a nuclear receptor and not a membrane receptor, this statement is a bit confusing. Also, the cited reference only comments on membrane vesicles in the context of adrenocortical cells and increased release of steroids from these cells under stress, which is not applicable to the HEK239T cells. Please clarify and/or change references.

Response: We sincerely thank the reviewer for this insightful comment and fully agree with the point raised. However, following the implementation of a new experimental design and updated analytical methods, this specific result is no longer included in the revised manuscript.

Reviewer #2

The study by Zhang and colleagues addressed an important question regarding how HEK293 endogenous rhythms function. The authors highlighted the consequences of not considering time in experiments, which is highly relevant to the field. However, upon careful review of the methods, the current rhythm analyses are underpowered and susceptible to a high rate of false positives. The authors used uncorrected p-values to determine rhythmicity, resulting in a large number of rhythmic genes—close to half of the transcriptome. Based on this data, they also performed phase set enrichment analyses (PSEA), which similarly used uncorrected p-values. Overall, the manuscript’s conclusions are not supported by robust statistical analysis. Although the manuscript is well-written and presents a good idea, the conclusions drawn are insufficient. The authors should reanalyze their data while accounting for false discovery rates, consider using sine-wave-based methods to accurately estimate phase, and then perform PSEA. Additionally, they could use DEG analyses to identify truly arrhythmic processes, providing a balanced view of major processes that are either under temporal control or not. This could become a valuable resource for the scientific community. I also provide some other comments below.

Response: Thank you very much for your insightful and constructive comments regarding our manuscript. We have carefully revised the statistical analyses and updated our findings in accordance with your suggestions.

Introduction

Add original studies to support the claims. For instance, reference 6 is a review. The authors should cite the original paper.

Response: We thank the reviewer for raising this point. In the revised manuscript, we have cited the original papers. Specifically, Reference 6 has been replaced with the relevant article.

M&M

Usually, after dexamethasone synchronization, one should wait 24 hours to avoid the immediate GR-induced effects. In this setup, the authors proceed immediately after synchronization, which should be disclosed as a limitation. Additionally, the authors could select a few relevant genes and repeat the experiments, this time waiting 24 hours, to show that the induced effect is clock-mediated rather than GR-induced.

Response: We thank the reviewer for pointing this out and apologize for the experimental design mistake. As both the Editor and the Reviewer #1 also pointed out, we have redesigned the experiment to provide a more comprehensive temporal analysis. Specifically, we collected samples at 13 time points over a 48-hour period, with sampling initiated 24 hours after DEX synchronization (Line 132-136). The process of sample collection and data analysis is shown in Fig 1A.

The authors used only the metacycle package to estimate rhythmicity, which includes ARSER, JTK_Cycle, Lomb-Scargle, and meta2d. In this setup, it is likely that Meta2D incorporated JTK_Cycle and Lomb-Scargle methods. An important limitation of this approach is that these methods are not sine-wave based, making phase and amplitude estimation difficult. Specifically, the authors used the calculated phase for PSEA analysis. Importantly, using a sine-wave method in addition could help obtain high-resolution phase estimates, although the overlap between detection methods is often quite limited.

Response: We sincerely thank the reviewer for this constructive suggestion regarding rhythmicity analysis. In the revised manuscript, while we use the MetaCycle package for initial rhythmicity estimation due to its effective integration of multiple algorithms, we have additionally employed a sine-wave-based least-squares method to estimate of phase and amplitude. The code used for this analysis has been provided in the S1 file and S2 file.

In addition, all analyses were based on a standard p-value < 0.05, which, considering RNAseq data, could inflate the rhythm detection. In this case, a false-discovery rate is necessary. Since some of these methods have low statistical power, using a p-value < 0.01 could help reduce false positives. Not surprisingly, the authors report almost half of the transcriptome as rhythmic, which is likely due to the accumulation of false positives.

Response: We thank the reviewer for pointing out this. To minimize false positives, we have implemented more stringent statistical criteria in the revised manuscript. Specifically, rhythmicity was determined using Benjamini-Hochberg adjusted p-values (BH.Q < 0.05) within the MetaCycle pipeline, which significantly reduced the number of identified rhythmic genes (Figure 1D). Furthermore, a false-discovery rate (FDR) was applied to the pathway enrichment analysis, and PSEA results were refined using the adjusted Kuiper q-value.

The same issue applies to the PSEA analysis, as the authors used a p-value < 0.05. Here, a q-value should be used instead of < 0.05 to reduce false positives. Data processing for PSEA is unclear and not reproducible. This method produces a large number of biological processes, often overlapping. How was the merging or grouping of similar processes done? Based on the data shown in fig. 2I, this reviewer cannot replicate these findings with the information provided by the authors. Importantly, PSEA data is not included in the materials; this must be provided.

Response: Thanks for bringing this point up. The Kuiper q-value was applied to the PSEA analysis to reduce false positives. We apologize that we have not yet identified a robust methodology for the merging or grouping of similar biological processes. We would be highly appreciative of any specific suggestions the reviewer might have and would be eager to implement them. In the revised manuscript, we observed that the phases of rhythmic genes are relatively concentrated. Consequently, the PSEA results revealed processes consistent with our pathway enrichment analysis, with these processes predominantly clustered between T12 and T14. The PSEA data have been included in the S5 File, which has been uploaded as Supplementary Material, and the detailed PSEA workflow is provided in the S3 File.

Results

The authors should consider using a sine-wave baseline method for the clock genes to better assess the classic anti-phase relationship. Surprisingly, DBP, a high-amplitude gene, is not rhythmic.

Response: In the revised manuscript, we integrated MetaCycle with a sine-wave-based least-squares method to analyze the rhythmicity data. Upon analyzing the 28-hour dataset using stringent Benjamini-Hochberg (BH) adjusted p-values (BH.Q < 0.05), we observed that several core clock genes, including DBP, did not exhibit significant rhythmicity in the current sample set.

The phase shown in D could be better represented with a rose plot.

Response: As the core clock genes lacked significant rhythmicity in the new dataset, we did not include their phase informa

---

## [Editor Report · Decision Letter 1]

16 Feb 2026

Temporal dynamics and functional annotation of transcriptome rhythmicity in HEK293T cells

PONE-D-25-55405R1

Dear Dr. Zhang,

We’re pleased to inform you that your manuscript has been judged scientifically suitable for publication and will be formally accepted for publication once it meets all outstanding technical requirements.

Kind regards,

Nicholas S. Foulkes, D.Phil

Academic Editor

PLOS One
---

## [Editor Report · Acceptance letter]

PONE-D-25-55405R1

PLOS One

Dear Dr. Zhang,

I'm pleased to inform you that your manuscript has been deemed suitable for publication in PLOS One. Congratulations! Your manuscript is now being handed over to our production team.

Kind regards,

on behalf of

Dr. Nicholas S. Foulkes

Academic Editor

PLOS One